# Recycled Waste as Polyurethane Additives or Fillers: Mini-Review

**DOI:** 10.3390/ma17051013

**Published:** 2024-02-22

**Authors:** Edyta Pęczek, Renata Pamuła, Andrzej Białowiec

**Affiliations:** 1Department of Applied Bioeconomy, Wrocław University of Environmental and Life Sciences, 37a Chełmońskiego Str., 51-630 Wrocław, Poland; edyta.peczek@upwr.edu.pl; 2Selena Industrial Technologies Sp. z o.o., Pieszycka 3, 58-200 Dzierżoniów, Poland; renata.pamula@selena.com

**Keywords:** polyurethane adhesives, circular economy, recycling, organic waste, mineral waste

## Abstract

The intensive development of the polyurethanes industry and limited resources (also due to the current geopolitical situation) of the raw materials used so far force the search for new solutions to maintain high economic development. Implementing the principles of a circular economy is an approach aimed at reducing the consumption of natural resources in PU production. This is understood as a method of recovery, including recycling, in which waste is processed into PU, and then re-used and placed on the market in the form of finished sustainable products. The effective use of waste is one of the attributes of the modern economy. Around the world, new ways to process or use recycled materials for polyurethane production are investigated. That is why innovative research is so important, in which development may change the existing thinking about the form of waste recovery. The paper presents the possibilities of recycling waste (such as biochar, bagasse, waste lignin, residual algal cellulose, residual pineapple cellulose, walnut shells, silanized walnut shells, basalt waste, eggshells, chicken feathers, turkey feathers, fiber, fly ash, wood flour, buffing dust, thermoplastic elastomers, thermoplastic polyurethane, ground corncake, *Tetra Pak^®^*, coffee grounds, pine seed shells, yerba mate, the bark of Western Red Cedar, coconut husk ash, cuttlebone, glass fibers and mussel shell) as additives or fillers in the formulation of polyurethanes, which can partially or completely replace petrochemical raw materials. Numerous examples of waste applications of one-component polyurethanes have been given. A new unexplored niche for the research on waste recycling for the production of two components has been identified.

## 1. Introduction

The first polyurethane (PU) was synthesized by Dr. Otto Bayer and his team in 1937 at IG Farbenindustries by reacting a polyester diol with a diisocyanate [1]. They developed a new polyisocyanate polyaddition process [2]. This discovery opened a new path in macromolecular chemistry and completely revolutionized the polymer and coatings industry [3,4]. The potential of this polyurethane was noticed during the Second World War when it was used for the impregnation of paper and the production of mustard gas-resistant clothing, high-gloss airplane finishes, chemical, and corrosion-resistant coatings to protect metal, wood, and masonry. With a relatively short history of just over 86 years, polyurethanes have become one of the most dynamic groups of polymers, and their field of application covers almost all areas of polymer use (Figure 1) [5,6,7]. The most common PU application is used to produce flexible foams and rigid foams.

Nowadays, polyurethanes are used in almost all aspects of daily life, changing the quality of people’s lives. Cushion materials, furniture, automotive, bedding, seating cars, shoe soles, medical devices, wood substitutes, electronics, and packaging, are only a few common examples of polyurethane used in everyday life (Figure 2) [8,9].

It is expected that demand for polyurethane connected with building insulation will increase during the forecast period because of sustainability concerns. Sustainability in construction development is a broad area and involves several steps that need to be taken in the early stages of construction due to the high potential environmental impact. The global market is segmented by the presence of major players operating in different regions of the world, as well as various small and medium-sized regional and national players. Polyurethane manufacturers include both large and medium-sized companies operating on a global or regional scale. Regional players have strong distribution networks and are prepared to modify their products in line with regulatory changes at the regional and national levels. The largest share in the global PU market in 2021 was held by Asia at 45%, followed by North America at 26%, and Europe at 19% [10].

Most of the market players compete based on raw materials such as polyols, methylene diphenyl diisocyanate, and toluene diisocyanate employed in the production of polyurethanes. The prominent players in the global Polyurethane market are Dow Inc.(USA), BASF SE (Germany), Covestro AG (Germany), Huntsman International LLC (USA), DIC Corporation (Austria), Eastman Chemical Company (USA), Mitsui & Co. Plastics Ltd. (USA), Mitsubishi Chemical Corporation (Japan), Recticel NV/SA (Belgium), The Lubrizol Corporation (USA), Woodbridge (Canada), RTP Company (USA), RAMPF Holding GmbH & Co. KG (Germany), and Tosoh Corporation (Japan) [10,11,12].

## 2. Polyurethane Classification, Structure and Properties

As the name suggests, “polyurethane” (PU) describes a huge group of different polymers. Depending on the starting materials, the final product will have different types, properties, and uses. They have only the same urethane groups and the other parts of their chemical structure are completely different, so the form, structure, and, above all, properties of polyurethanes differ from each other. The characteristics of the polyurethanes depend on their structure and the combination of hard segments and soft segments or different stoichiometric ratios (NCO:OH) [6,13,14,15]. PU is a polymer formed by the reaction of OH (hydroxyl) groups of polyols with NCO (isocyanate functional) groups of isocyanates. This reaction is exothermic and leads to the formation of urethane groups [15].

The basic raw materials for obtaining PU are isocyanates, oligomerols (called macropolyols, oligopolyols, or polyols) with a long elastic chain, chain extenders (glycols and diamines); additionally, they may include catalysts, additives, antioxidants, fillers, fiber, drying agents and adhesion promoters [16,17,18].

Polyols are polyhydric compounds that, depending on the introduced group, ester or ether, are divided into oligoesterols and oligoetherols. Oligomerols are compounds with long, flexible chains. They finish with at least two hydroxyl groups with a molecular weight of 1000–18,000 [19]. They usually represent about 2/3 of the PU composition. They give PU elasticity and softness, resistance to low temperatures.

Isocyanates are chemical compounds containing the -N=C=O grouping, in which the binding atom is a nitrogen atom [20]. The classification of isocyanates is presented using Figure 3.

In the synthesis of polyurethane materials, the most commonly used isocyanates are diphenylmethane diisocyanate (MDI) and toluene diisocyanate (TDI). Slightly less common are aliphatic isocyanates such as hexamethylene diisocyanate (HDI), and isophorone diisocyanate (IPDI). Aromatic isocyanates are typically used in polyurethane adhesives due to their reactivity and lower cost of formulation. However, aromatic isocyanates are not as lightfast and not as oxidation-resistant as aliphatic isocyanates [16,17,18].

Extenders are bifunctional compounds, such as diols and diamines. These are small-molecule compounds that increase the size of rigid segments and the density of hydrogen bonds as well as the molecular weight of PU [17,18].

Fillers are substances that do not mix homogeneously with PU components. They reduce the cost of PU and improve some of its properties. They are divided into powder, fiber, and flake fillers. The best known are calcium carbonate, barium sulfate, silica, talc, and so on [17,18].

The catalysts for the synthesis of polyurethanes are tertiary amines, trialkylphosphines, salts of certain metals (Bi, Fe, Sn, Zn), and organotin compounds (e.g., tin 2-ethylcapronate and dibutyl dilaurate). The catalysts regulate specific reactions and control the rate of chain growth, foaming and curing of the polyurethane. They also enable PU to obtain optimal properties (at a given composition) at economically justified rates [16,17,18,20,21,22,23].

In recent years, there has been growing interest and a developing trend among companies, researchers, and institutions to start using waste raw materials. This development trend is influenced by several key factors; the increasing environmental awareness of society and pressure for more sustainable development are prompting companies to seek alternative sources of raw materials. Waste raw materials are an attractive option because their use contributes to reducing negative environmental impacts. Waste raw materials are an important part of this concept as they enable recovery, recycling, and reuse. Technological advances allow raw material waste to be processed more efficiently, making it a more attractive source of raw materials for various industries. Modern technologies allow waste raw materials to be segregated, processed, and used more efficiently. In addition, waste raw materials are often cheaper than virgin raw materials, which attracts companies that can reduce production costs, or even gain additional revenues by using them. All these factors make waste raw materials an attractive and promising source of raw materials for many sectors of the economy. The growing trend of using waste raw materials has the potential to reduce raw material waste, reduce environmental pressure, and promote sustainable development [24,25,26].

## 3. Circular Economy Principles

The Industrial Revolution has brought with it extravagant innovations that shape every aspect of our lives today. The world has witnessed a massive increase in population, which is leading to the depletion of natural resources and an increase in environmental pollution, coupled with a higher standard of living [27]. The fear of the depletion of oil reserves and the capricious course of raw materials is mobilizing researchers/scientists to look for new solutions that will make it possible to replace mineral or petrochemical fillers with mineral or natural waste (meaning solid waste), thus becoming a part of the Circular Economy strategy.

A circular economy is a business model that minimizes the consumption of raw materials and the generation of waste. The goal is to reduce greenhouse gas emissions and energy consumption (Figure 4). It creates a closed process cycle in which the waste generated is treated as a raw material in subsequent production steps. Such policies can help create a sustainable, low-carbon, and competitive economy. Implementing changes to a circular economy not only helps the environment, but also differentiates the company from the market, reduces operating costs, and improves the quality of services provided.

The circular economy is associated with deep changes in the production and consumption chain and the restructuring of the industrial system. This strategy envisages a transition from a linear model based on the production (take)—consumption (make)—dispose scheme to a loop model in which waste is used as a raw material. In this solution, the circular economy, thanks to good supply chain management, eliminates the concept of the end of product life.

The circular economy has been one of the main directions of strategic thinking about the development of the European Union economy since the announcement by the European Commission of the Circular Economy Action Plan in 2015. Since then, the Commission has published several new directives on its aspects, the last one in 2020. It is part of the European Green Deal and aims to achieve climate neutrality and resource efficiency by 2050 [28,29,30,31].

As part of sustainable development and a circular economy approach, an increasing number of companies and organizations are experimenting with recycling or using raw material waste to produce polyurethanes. Thanks to this innovative perspective, we can reduce the burden on the natural environment and promote more sustainable and environmentally eco-friendly solutions in the adhesive industry.

## 4. Possibilities of Waste Recycling in Polyurethanes

Waste generation has not decreased despite efforts from the EU and on a national level. The EU generates 2.5 billion tons of waste from all economic activities, or 5 tons per inhabitant per year, with an average of almost half a ton of municipal waste per inhabitant. Decoupling waste generation from economic growth requires intensive efforts along the entire value chain and in every household [30].

In 2020, when it comes to only two-component spray polyurethane foam (SPF) about 800,000 tons were produced, where additives could account for up to 25% of the formulation [31]. The following question arises, could we replace the additives or fillers with waste? It turns out that many scientists have already taken up this challenge. There are many papers where, apart from biopolyols, polyurethane foams were produced with the application of a different kind of waste. Table 1 shows the possibilities of recycling waste in polyurethanes.

After conducting the classification based on the utilized waste and the obtained polyurethane materials (PUR), it becomes apparent that there is significant potential to utilize waste in the production of various polyurethane products. In order to classify the waste in accordance with the Commission’s guidelines on the technical classification of waste and Commission Decision of 18 December 2014 amending Decision 2000/532/EC on the List of Waste pursuant to Directive 2008/98/EC of the European Parliament and of the Council (2014/955/EU), it can be divided into several main categories: wastes resulting from exploration, mining, quarrying, the physical and chemical treatment of minerals (basalt waste); waste from agriculture, horticulture, aquaculture, forestry, hunting and fishing, food preparation and processing (bagasse, biochar, sawdust, coffee grounds, residual pineapple cellulose, chicken feathers, egg shells, turkey feather fiber, cuttlebone, ground corncake, sunflower husks, rice husks, and buckwheat hulls, pine seed shells, yerba mate, walnut shells, silanized walnut shells, mussel shell); waste from wood processing and the production of panels and furniture, pulp, paper and cardboard (wood flour, buffing dust); waste from organic chemical processes (waste lignin); waste not otherwise specified in the list (thermoplastic elastomers and thermoplastic polyurethane, *Tetra Pak^®^,* glass fibers, ground rubber powder, tire rubber, glass fibers) [83]. As indicated in Table 1, these waste materials can lead to a significant improvement in the mechanical, thermal, and thermomechanical properties of the obtained polyurethane foams, coatings, sealants, TPU, elastomers, and adhesives. However, before we use waste materials, it is important to consider several key factors. First and foremost, it is essential to ensure that these materials are safe for both human health and the natural environment. Therefore, thorough processing and purification are necessary before their utilization to prevent potential contaminants that could adversely affect the final product. Secondly, the selection of appropriate waste materials should be made considering their compatibility with the production process and their ability to provide the desired properties to the final product. These materials should meet specific technical and quality requirements to avoid jeopardizing product quality. Finally, equally important is the optimization of the quantity of waste materials used in formulations and their availability on the market. The number of materials should be adjusted to production needs to minimize waste generation while maintaining product quality and performance. It is also crucial to note that such waste utilization processes contribute to reducing the amount of waste directed to landfills, supporting the concept of a circular economy. Through the meticulous classification and analysis of the utilized waste and its impact on the properties of PUR materials, a better understanding of the recycling potential and sustainable utilization of resources in the polyurethane industry can be achieved. Furthermore, this practice can contribute to raising awareness of environmental issues within society and promoting more sustainable industrial practices [14,15].

As can be observed from Table 1, scientists focused predominantly on the potential of using waste raw materials in the production of polyurethane foams because there is a huge potential for recycling and sustainable waste management in this area. Polyurethane foams account for 65% of the total polyurethane market, with versatile applications covering industries such as construction, automotive, furniture, and many others. Thanks to their excellent insulation, soundproofing, and comfort properties, polyurethane foams are present in many aspects of our daily lives, which contributes to the increased research interest in this field [6].

The examples mentioned refer to various types of polyurethanes, focusing mainly on polyurethane foams. Nevertheless, there is another category of polyurethanes that often escapes the attention of researchers, namely two-component polyurethane adhesives, on which there is a limited availability of scientific publications.

It is also worth noting that the market for filled two-component polyurethane adhesives is relatively niche, which opens up the opportunity to differentiate product offerings and reach new customer groups [84]. Exploring the area of filled two-component polyurethane adhesives is, therefore, an exciting prospect. These adhesives have unique properties and performance advantages over their single-component counterparts. By adding fillers, such as nano-powders, fibers, or other additives to the adhesive matrix, manufacturers can tailor the properties to meet specific application requirements [14].

It is worth noting that the global polyurethane adhesives market reached a value of $18.47 billion in 2021 and is projected to reach $37.62 billion by 2028, with an average annual growth rate of 9.3% [24]. Furthermore, these polyurethane adhesive systems can incorporate up to 60% organic fillers, making them, in our opinion, an excellent candidate for the utilization of raw material waste. It is important to highlight that many of these waste materials have already been proven successful in the context of polyurethane foams, coatings, sealants, TPU, and elastomers. Although we have not come across existing research in this specific domain during the literature review, we believe it is worth initiating investigations in this direction. The two-component PU may increase the potential for waste recycling, as it allows to recycling synergistically two types of waste. This opens a new research niche for further investigations and technology developments.

## 5. Adverse Effects of Using Waste Materials—Critical Discussion

It is worth noting the potential negative consequences associated with the use of waste materials as fillers in the polyurethane industry.

If the waste contains pollutants, there is a risk of contaminating the structures of buildings. These substances can permeate through building materials, adversely affecting the durability and integrity of the structure. Such contaminants may lead to the degradation of materials, resulting in a decline in the quality of the structure. Contaminated buildings with harmful substances may pose a risk to the health of residents. Inhaling air polluted with these substances can lead to health problems, especially if they are hazardous substances. Respiratory issues, skin problems, or even impacts on the nervous system are possible with prolonged exposure. Harmful substances present in waste materials, when used as fillers in polyurethane adhesives, can affect indoor air quality. Emitting toxic chemical compounds into the air can lead to indoor air pollution, which is particularly significant for individuals spending extended periods indoors. Harmful substances may cause allergic reactions or skin diseases in individuals exposed to them over an extended period. Contact with contaminated structures or adhesives releasing toxins can cause skin irritation, skin allergies, or other dermatological problems [85]. Properties exposed to the negative effects associated with the use of waste materials in polyurethane products may be less attractive to investors or potential buyers. Therefore, the manufacturers of polyurethane adhesives and construction designers need to be aware of the potential hazards associated with the use of waste materials, apply precautions, and employ technologies aimed at minimizing these adverse effects on buildings and residents’ health. In Poland, numerous building regulations establish detailed requirements and standards in various areas, such as building materials, insulation, electrical installations, ventilation, etc. Additionally, many companies specializing in the construction industry recommend using products with A+ class certificates, CE markings, or construction markings B. Products with the EcoLabel ecological mark are particularly popular because there is a strive for buildings to obtain certifications such as LEED (Leadership in Energy and Environmental Design), BREEAM (Building Research Establishment Environmental Assessment Method), DGNB, HQE (Haute Qualité Environmentale), or WELL [86,87].

In addition, there is a risk of lowering the mechanical performance of the final polyurethane product, which should be taken into account when using waste raw materials as fillers. These raw materials can have a variety of physical and chemical properties which can affect the strength, flexibility, and overall stability of the polyurethane structure. Lower-quality waste raw materials can reduce the mechanical strength of polyurethane, especially when used as fillers. This can lead to a reduction in the structure’s ability to withstand tensile forces, a key parameter for materials used in load-bearing structures. Waste raw materials, when used in excessive quantities or of low quality, can affect the resistance of polyurethane to compressive forces. This phenomenon can increase the risk of structural deformation and lead to a loss of structural stability. In the case of unsuitable waste raw materials, the elasticity of polyurethane can be negatively affected. This can have consequences for the material’s behavior under dynamic conditions, especially for structures subjected to movement, dynamic loads, or temperature changes. Waste raw materials used in the manufacturing process can also affect the adhesive properties of polyurethane. This, in turn, can lead to the weakening of the connections between different structural components, which poses a direct threat to the durability and stability of the entire structure. If waste materials reduce the strength and stability of a structure, this can limit its ability to carry loads. This can be a problem, especially for load-bearing structures such as beams, columns, or other structural elements. Low-quality waste raw materials can affect the resistance of the finished polyurethane product to weathering or temperature changes. This phenomenon can shorten the life of the structure and require more frequent repairs or replacements. All of these factors combined can lead to a reduction in the overall durability of the structure, which is particularly important when using polyurethane in building, transportation, or other structures where high mechanical stability is required. In the literature, there is already a wealth of information discussing the deterioration of parameters, including a decrease in compressive strength, associated with the excessive use of fly ash and cement kiln dust. This is attributed to the adverse impact on the cell morphology of polyurethane foam, which constitutes a significant factor determining these unfavorable changes. In the available literature, there are numerous reports on the deterioration of parameters, particularly a decrease in compressive strength, associated with the excessive use of fly ash and cement kiln dust. This adverse effect is mainly attributed to changes in the morphology of polyurethane foam cells, which is a key factor determining these negative alterations [88]. Kurańska et al. also mentioned a decrease in the strength parameter in their article, conducting tests using basalt waste [89]. It is also worth noting that excessive amounts of hazelnut, cellulose, chitin, and eggshell negatively affect the split tear strength [42,90].

Incorporating waste raw materials into the polyurethane production process presents some challenges that may require additional resources and expenditures. In the context of technology adaptation, it may be necessary to redesign the production line or adjust process parameters to take into account the specific characteristics of waste raw materials. Appropriate processing and blending technology must be adapted to ensure the consistent and stable quality of the final product. Quality control becomes a key aspect when introducing waste raw materials. Due to their diverse chemical, physical, and qualitative nature, it is necessary to establish more precise inspection, monitoring, and testing procedures. This acts not only as a safeguard against instabilities in the product but also as a preventive measure for both environmental and health risks to workers and end users. These additional measures, such as new technological processes, increased quality control, and more advanced testing procedures, can lead to increased production costs. Investments in new technologies, employee training, and the purchase of advanced equipment are inevitable, and their introduction can affect operational efficiency and generate additional operating costs.

Polyurethanes, constituting a fundamental component of numerous everyday consumer products, occupy a pivotal role within contemporary industrial landscapes. Notwithstanding their multifaceted utility and widespread appeal, these substances also pose substantial threats to human health. Among the principal risk factors attributed to polyurethanes lies the presence of isocyanates, reactive chemical compounds fraught with peril to human well-being. Moreover, polyurethanes often incorporate solvents and other additives, thereby compounding the potential hazards to human health. While airborne exposure to isocyanates has historically garnered significant attention as a primary concern, the significance of cutaneous exposure in sensitization processes and the pathogenesis of isocyanate-induced asthma is increasingly underscored. Indeed, dermal contact with isocyanates can incite sensitization cascades, precipitating allergic manifestations upon subsequent exposures. Anecdotal testimonies and empirical investigations substantiate the notion that dermal exposure possesses the capacity to elicit systemic sensitization, thereby contributing to the genesis of asthma even in contexts characterized by diminished airborne concentrations. Regulatory frameworks governing cutaneous exposure to chemical agents notably lag behind their counterparts overseeing inhalation routes. Conscientious acknowledgment of the latent risks associated with dermal exposure to isocyanates can furnish regulatory bodies and occupational health authorities with critical insights aimed at fortifying safeguards for both laborers and consumers. The imperative to deploy a synergistic array of mitigation strategies encompassing engineering controls, isocyanate substitution protocols, personal protective equipment (PPE), procedural adjustments, and educational initiatives, is indispensable for curbing both aerial and cutaneous exposures within work environments. Occupational asthma within the European Union predominantly emanates from exposures to diisocyanates, particularly exemplified by methylene diphenyl diisocyanate (MDI), toluene diisocyanate (TDI), and hexamethylene diisocyanate (HDI). These potent sensitizers of the skin and respiratory system find pervasive usage across sectors encompassing coatings, adhesives, sealants, elastomers, and the production of polyurethane. Regulatory interventions, including stringent limitations on diisocyanate concentrations within products and mandatory instruction on risk management protocols, are directed toward curtailing occupational exposures and ameliorating the prevalence of diisocyanate-related asthma cases. To address the hazards concomitant with diisocyanate-induced asthma, regulatory mandates, such as the prescription of products exceeding 0.1% by weight diisocyanate content, have been enacted under the auspices of the Registration, Evaluation, Authorization, and Restriction of Chemicals (REACH) Regulation within the European Union. This measure endeavors to abate occupational encounters with diisocyanates and attenuate the frequency of diisocyanate-induced asthma occurrences across the European Union [91,92].

However, it is worth noting that despite the short-term financial challenges, the long-term benefits of using waste raw materials can include aspects of sustainability, a positive impact on a company’s reputation, and responding to growing consumer expectations of going green and recycling. In addition, technological advancements and innovations in the processing of waste raw materials can, in the long term, help reduce costs and improve process efficiency.

Accordingly, the industry should focus on research and innovation aimed at minimizing the negative impacts associated with the use of waste raw materials. Implementing sustainable and responsible manufacturing practices is key to minimizing environmental impact and maintaining high product quality. By actively engaging in research, innovation, and implementing responsible practices, the industry can contribute to a more sustainable and environmentally friendly manufacturing sector. This approach not only minimizes the negative effects of using waste raw materials, but also makes positive changes in terms of ecology and product quality.

## 6. Conclusions

In today’s world where the demand for green industries is extremely high, polyurethane products are also gaining support in countries around the world and are rapidly developing as new building materials. The presented results of the work of many authors indicate that much has already been conducted to make polyurethanes more environmentally friendly materials. A comprehensive discussion of all the interesting work would be beyond the scope of this review. Nevertheless, the significant potential of the shift from petrochemical fillers/additives to renewable resources (which are currently useless waste) in the very large sector of polyurethane polymers has been demonstrated. In particular, it shows a new direction of their application, which are two-component polyurethane adhesives. It is high time we stopped looking at waste as “garbage”, that it is something superfluous, useless, or inconvenient. Waste becomes valuable secondary raw materials that can be recycled to give them a second life in the same or different form. This approach leads, additionally, to better protection of the natural environment and develops a modern circular economy in which almost nothing is wasted and raw materials and items are reused many times. In the upcoming article, I intend to focus on the selection of raw materials that could be suitable as fillers for two-component polyurethane adhesives, as well as evaluating their market availability.

## Figures and Tables

**Figure 1 materials-17-01013-f001:**
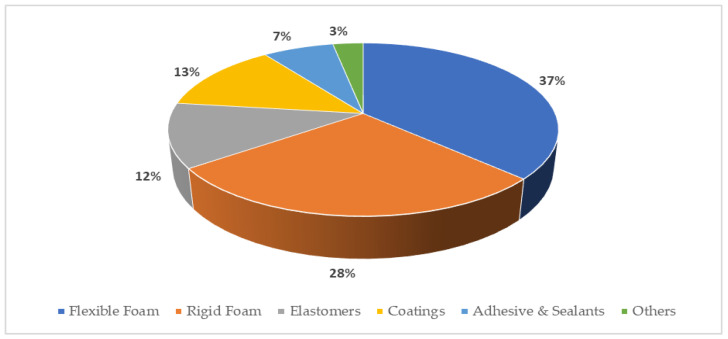
Uses polyurethanes in global markets [7].

**Figure 2 materials-17-01013-f002:**
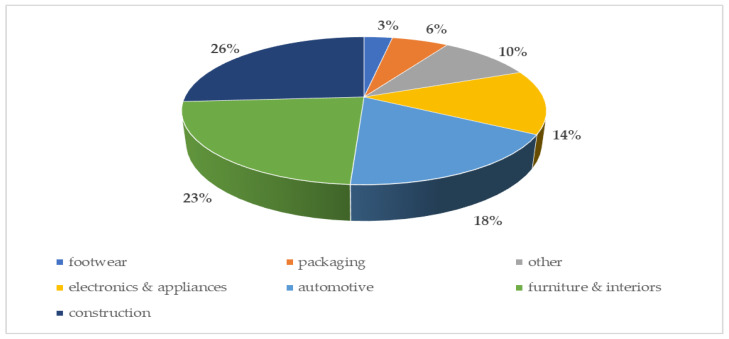
Polyurethane market share by application 2021 [%] [10].

**Figure 3 materials-17-01013-f003:**
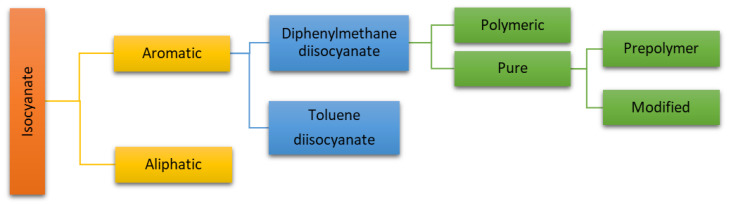
The classification of isocyanate [17].

**Figure 4 materials-17-01013-f004:**
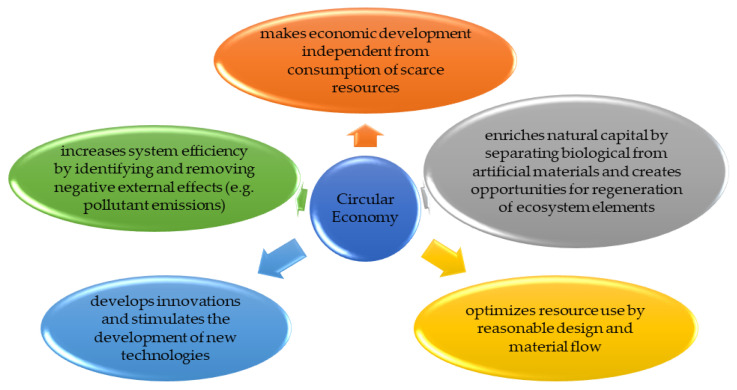
The main goals of a circular economy [27,28,29,30,31].

**Table 1 materials-17-01013-t001:** Opportunities for Utilizing Waste in Polyurethanes.

Waste	Product	Max Amount of Filler [%]	Advantages of Using Waste
biochar [32]	rigid polyurethane foams (RPUF)	20	- improves dimensional and thermal stability - increase in the proportion of the biochar reduces their compressive strengths at 10% strain and the stiffness of the materials
bagasse [33]	rigid polyurethane foams (RPUF)	12.5	- significantly changes the physical structure (cellular structure) - increase the number of hard segments, improve density and compressive strength - improves RPUF biodegradation - as the bagasse content increases, the value of λ increases
waste **lignin** [34,35,36,37]	polyurethane foams	20	- decrease in the hydrophobicity of the foams - increase the density of the foam - improve the mechanical properties
residual algal cellulose [38]	rigid polyurethane foam	1.6	- increase the density of the foam, - reduces the mechanical resistance of the module to compression
residual pineapple cellulose [39]	rigid polyurethane foam	5	- as the amount of cellulose fibers increases, the compression density and the apparent density decreases - reduces foam stiffness
walnut shells/silanized walnut shells [40,41]	polyurethane foam	5	- prolongs foaming time - influences the cellular structure of PU foams - results in a more homogeneous structure - reduces cell size
basalt waste [42]	rigid polyurethane foam	40	- increase the reactivity of the polyurethane system - improves thermal stability - worsens chemical properties
eggshells [43,44,45,46,47]	polyurethane foam	25	- increases apparent density, - suitability for the Cosmetics Industry - positively influenced the mechanical properties, particularly in terms of elongation and tensile strength
chicken feathers [48,49]	rigid polyurethane foams	15	- reduces the compression strength - improves thermal insulation properties
turkey feather fiber [50,51]	rigid polyurethane foams	15	- alters thermal conductivity, leading to improved insulation properties - the excessive incorporation of fibers decreases air permeability by creating a dominant barrier effect, which restricts airflow and reduces perforations in the foam - at higher concentrations, the cellular morphology exhibited distortion and irregularity
fly ash [52,53,54,55,56,57]	rigid polyurethane foams	15	- improve compressive strength and Young’s modulus - enhance mechanical properties, thermal stability, and thermal properties - impairs the brittleness
wood flour [58,59,60,61]	rigid polyurethane foams	15	- with the increase in the concentration of wood flour, there is a decrease in compression properties and a slight increase in thermal conductivity - improves thermal stability
buffing dust [62]	rigid polyurethane foams	5	- improve the mechanical properties - reduce the transition glass temperature
thermoplastic elastomers and thermoplastic polyurethane [63]	rigid polyurethane foams	2.5	- increase the compression modulus, tear resistance, and Young’s Modulus
ground corncake [64]	flexible polyurethane foams	10	- improve the mechanical strength of the foams, resulting in better tensile strength and elongation at break values - enhance the thermal behavior of the foams, such as thermal conductivity and temperature response - improve the morphology and cell shape of the material
*Tetra Pak^®^* [65]	polyurethane foams	20	- led to improved thermal stability, especially during the first degradation stages - a significant increase in compressive stress, with values at 10% deformation four times higher than that of pristine foam
coffee grounds [66]	rigid polyurethane foams	20	- exhibit better mechanical properties, particularly in terms of compressive strength - has no notable influence on the fire resistance - enhance thermal conductivity
sunflower husks, rice husks and buckwheat hulls [67,68,69]	rigid polyu-re-thane foams	15	- improve higher compressive strength - 5 wt% of rice hulls showing the highest resistance to fire - lower water absorption - higher apparent density
pine seed shells [70]	polyurethane foams	15	- lead to improvements in thermal properties, mechanical stability, and insulation properties
yerba mate [70]	polyurethane foams	15	- enhance mechanical strength and thermal properties
sawdust [71]	polyurethane foams	20	- reduce brittleness and maintain high dimensional stability - reduce thermal conductivity - causes a slight decrease in compressive strength - decrease in its bending strength and an increase in water absorption
potato protein [72]	rigid polyurethane foams	5	- improve compressive strength - low thermal conductivity - low water absorption
ground coffee [73]	polyurethane-polyisocyanurate	15	- lower brittleness and compressive strength - increase absorptivity and impregnability
waste tire rubber [74]	polyurethane foams	20	- increase in the density - decrease in cell size and increase in pore volume - improved the thermal stability
the bark of Western Red Cedar [75]	adhesives	20	- improve the bonding efficiency and the formation of the adhesive layer at the joint
waste lignin [76]	adhesive	10	- improve rheological properties and adhesion properties of the adhesive to metal-to-metal and wood-to-wood substrates
rice husk ash/coconut husk ash [77]	elastomeric PU	10	- enhancing the overall adhesive properties of the composite
cellulose [77]	elastomeric PU	5	- leads to a decrease in the degree of phase separation within the composite material - improving thermal properties
eggshell [78]	waterborne polyurethane coatings	20	- significantly impacts the coating’s performance in terms of corrosion resistance - improve physical properties when exposed to artificial seawater - help maintain the morphological integrity of the coatings.
cuttlebone [79]	waterborne polyurethane coatings	7	- improvement in thermal stability, higher stiffness, and strength without sacrificing breaking elongation
glass fibers [80]	polyurethane coatings	60	- led to an improvement in the mechanical properties (higher tensile strength) - increase in flexural strength after exposure to aggressive chemicals - exhibit stronger adhesion to the concrete surface, better coating hardness, and lower water absorption
mussel shell [81]	TPU	30	- improve mechanical, thermal, thermo-mechanical, wear resistance, water absorption, and biodegradation properties
ground rubber powder [82]	polyurethane sealant	25	- enhances its mechanical properties by increasing tensile strength, improving hardness, and reducing elongation - adding filler decreases the spaces between the polymer chains, indicating a high degree of homogeneity between the filler and polymer

## Data Availability

Not applicable.

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
