# Peer review of "Recycled Waste as Polyurethane Additives or Fillers: Mini-Review"

_materials, 2024, doi:10.3390/ma17051013_

Round 1

Reviewer 1 Report

Comments and Suggestions for Authors

The authors present an review of the use of polyurethane waste in different contexts. The authors mention the circular economy, perhaps in some cases it is too repetitive and perhaps they can delete some phrases that are repeated over and over again, I have highlighted some of them in yellow in the pdf so that you can consider them as an example.

Then the authors give a bibliographic review of the uses of various wastes and their possible use in the production of PU, which I think is very comprehensive and correct. 

Then the authors list the potential problems of using recycled materials, both in terms of human health due to the use of some harmful substances, and also the potential negative consequences of using some of the recycled materials. I liked it because they create a "conflict" in the review and give a warning, which they later resolve by stating that investment is needed to work on optimising recycling.

I would ask the authors to reflect this in their conclusions, it is a very important point. 

The structure of the article reflects a very well thought out AND-BUT-THEREFORE structure. 

I recommend the publication of this article in Materials after taking into account these general observations (excessive repetition of the circular economy concept) and the minor comments I have made in the attached pdf. 

Author Response

The point-by-point response to the reviewer’s comments is included in the attached file.

Reviewer 2 Report

Comments and Suggestions for Authors

Dear Sir,

The subject is interesting, the idea of controlling the mechanical properties of polyurethanes by introducing various additives and in the same time find a usefull solution for recycling and repurposing various types of wastes is a valuable one. However, the Title is somehow misleading for the content of the manuscript – a better solution would be, for example, “Recycled wastes as polyurethane additives” or any other more suggestive solution (at first glance, one could think that the manuscript is dealing with recycling polyurethane wastes).

Other drawbacks of the manuscript are the following:

- The Introduction is rather long, with many unnecessary information. In fact, paragraphs 1, 2 and 3 should be reunited and shortened to form the Introduction (history, classification and structures of polyurethanes are all well-known, as well as what CE means).

- For a review, many important papers pertaining to the subject have been omitted. Practically, for a review, less then half of the references are pertaining to the subject (references 30 to 55 from a total of 62 citations – ca. 40%). Thus, the authors should extend the literature screening so at least 60% of the references are related to the topic of including wastes of various origins into polyurethanes. And since the authors could not encompass all pertinent papers, they should use the syntagm “a short review” (instead of “a review”).

- The authors should put all references for wastes reuse as PUR additives in the same Table (not some in Table 2 and the rest in the text that followed) and next made some classifications based on: 1) waste used, organized by similarities, origins etc.; 2) PUR material obtained, organized by properties and by destination of use, purpose etc.

- Paragraph 6 is a plus, but the authors should mention that also exposure to PUR could present adverse effects (there are numerous published studies on the subject)

Other comments:

- why Table 2 is entitled “Table 2. Differences between linear and circular economy [26].”? This is the legend of Table 1.

- Fig. 5: “Toluene”

- what has reference [1] has to do with Bayer and the discovery of polyurethanes? Use the original references (e.g. I.G. Farben (Otto Bayer, Werner Siefken, Heinrich Rinke, L. Orthner, H. Schild), German Patent DRP 728981, A process for the production of polyurethanes and polyureas, 1937 or Otto Bayer, Das Di-Isocyanat-Polyadditionsverfahren (Polyurethane), Angew. Chem. 1947, 59, 257–272. https://doi.org/10.1002/ange.19470590901)

As a conclusion, the manuscript presents some interest, but is far from complete: at least 10-12 more pertinent references must be added, the Introduction paragraph (that will regroup paragraphs 1, 2 and 3) must be shortened by eliminating many informations that are common knowledge, the various additives and PURs obtained must be classified in various forms and a more suggestive Title must proposed. Therefore, a major revision is needed.

Comments on the Quality of English Language

Small adjustments/corrections are needed.

Author Response

(The authors gave the same response as above.)

Reviewer 3 Report

Comments and Suggestions for Authors

The topic of the research work and manuscript is really interesting and provides new information. However there are some issues to be addressed towards its quality improvement before publication. The title does not make clear what kind of wastes the manuscript refers to, and the abstract as well should clarify the kind of wastes that could be applied in polyurethane adhesives. In line 62, check the word "used", the meaning is not clear. In 103 and 106 lines, please provide a reference. In 117-133, a reference would be also necessary. Even till the chapter 3, the reader has not understand which waste the manuscript focuses on. In line 182, check the use of "or". In table 2, something is wrong with the caption? In line 189, you refer to researchers without providing a literature. The scientific names of species should be presented in italics. In 208 line, check the meaning in "were found". It would be a good idea to clarify which is the concept of choosing the specific refered/studied waste types, since there are numerous other wastes that has been used in a similar concept. In 348 line, please provide the relevant study of http://dx.doi.org/10.4067/S0718-221X2017005000008 to support this statement. In 349 line, improve the "it's" and "haven't" in 355 respectively. In 368 line, what do you mean "toxic"?provide some examples. In 395, the "'s" is not necessary. Line 443, in the "company's reputation" the s should be erased. A brief comment on the availability of these materials and the different materials amounts available worlwide or in at least in Europe.

Comments on the Quality of English Language

The English language use is acceptable and comprehensible.

Author Response

(The authors gave the same response as above.)

Reviewer 4 Report

Comments and Suggestions for Authors

The manuscript "The recycling of the waste in polyurethanes – a review" has an important and actual subject of the research field.

The work presents the possibilities of recycling waste as additives or fillers in the formulation of polyurethanes, which can partially or completely replace petrochemical raw materials.

The manuscript present interesting and useful data.

However, some little corrections can be considered:

- Figures 3 and 4 must be more readable.

- Table 1 can be presented in graphical design.

- The valuable references were lost (is a possibility to extend this section of the manuscript).

Author Response

(The authors gave the same response as above.)

Round 2

Reviewer 2 Report

Comments and Suggestions for Authors

Dear Sir,

The authors have improved their manuscript by adding some references and by clearing several aspects, as it was pointed out in the precedent review. Some minor corrections must though be made:

Fig. 3: “diisocyanate” (not “diisocyjanate”) and “Toluene” (not “Toulene”)

Otherwise, the manuscript can be accepted for publication.

Reviewer 3 Report

Comments and Suggestions for Authors

As I have checked the authors have implemented the proposed changes in the revised verion of manuscript towards the improvement of their work. Almost all the changes have been implemented and in my opinion, the manuscript is well-prepared and organized enough to be accepted for publication in this journal.

Comments on the Quality of English Language

The English language use is acceptable.

Author Response

Thank you for your diligence and involvement in proofreading our article. I am grateful for your valuable insights and constructive criticism. Improving the text is a priority for us, so I appreciate every suggestion that contributes to enhancing the quality of our material. Once again, thank you for your assistance.